# Highly luminescent carbazole-functionalized tris(tribromophenyl)methyl radicals with stable circularly polarized photoluminescence

Larissa Schöneburg [1,5], Markus Gross [1,5], Philipp Thielert [2,3], Julia Zolg[1,4], Mona E. Arnold [1], Philipp A. Schuster[1], Bernhard Putz[1], Sabine Richert [2,3] ✉ & Alexander J. C. Kuehne [1,4] ✉

Brominated trityl radicals form stable enantiomers and present promising candidates for molecular qubits in spintronic and quantum technologies. However, their inherently weak photoluminescence ($\phi < 3$ %) limits optical spin readout. Here, we report the synthesis of three carbazole-functionalized tris(2,4,6-tribromophenyl)methyl (TTBrM) radicals via Buchwald-Hartwig cross-coupling. These chiral open-shell emitters exhibit high photo-luminescence quantum yields of up to $\phi = 72$%, comparable to their chlori-nated analogues, with red-shifted emission ($\lambda_{PL} = 646$-$688$ nm) due to enhanced charge-transfer character. Enantiomeric resolution via chiral chro-matography yields stable atropisomers ($T_{rac} > 90$ °C) with strong circular dichroism and circularly polarized luminescence showing dissymmetry factors ($g_{abs}$, $g_{lum}$) of order $10^{-4}$. EPR confirms spin localization on the trityl unit ($g \approx 2.006$) with phase memory times $T_m$ of $1.5 - 1.6$ μs, while donor methyla-tion improves photostability ($t_{1/2}$ up to 49 min). These bright, chiral TTBrM derivatives offer a significant advance toward stable molecular qubits.

Triphenylmethyl or trityl radicals represent one of the few classes of stable molecules with unpaired (odd) electrons that exhibit photoluminescence[1–3]. Typically, the trityl radical is halogenated so that the hemispheres above and below the p-orbital are screened at the methyl-carbon, where the unpaired electron resides[4,5]. The halogena-tion forces the phenyl rings into a propeller conformation, such that the trityl radicals become chiral (atropisomerism). However, in the case of chlorination, the resolved enantiomeric tris(2,4,6-tri-chlorophenyl)methyl (TTM) radicals can convert from the left- to the right-handed propeller and vice-versa (see Fig. 1).

The thermal barrier for this racemization is so low, that it occurs at room temperature within minutes – just long enough to determine a dissymmetry factor for the circularly polarized photoluminescence (CPL) of $g_{lum} \approx 5 \times 10^{-4}$ [6]. Bromination yields the tris(2,4,6-tri-bromophenyl)methyl (TTBrM) radical, which is stable in its axial chirality up to about 60 °C, with similar $g_{lum}$ of $7 \times 10^{-4}$ (see Fig. 1)[7]. Unfortunately, TTM and TTBrM exhibit only small photoluminescence quantum yields of $\phi = 3.1 - 0.8$ %, because the molecules are $C_3$ sym-metric and radiative transitions are forbidden[8]. Nonetheless, TTBrM has been reported to exhibit long coherence times, making it an

[1]L. Schöneburg, M. Gross, J. Zolg, Dr. M. E. Arnold, Dr. P. A. Schuster, B. Putz, Prof. Dr. A.J.C. Kuehne OC III – Institute of Organic and Macromolecular Chemistry Ulm University Albert-Einstein-Allee 11, 89081 Ulm, Germany. [2]P. Thielert, Prof. Dr. S. Richert Institute of Physical Chemistry II Ulm University Liese-Meitner-Straße 16, 89081 Ulm, Germany. [3]P. Thielert, Prof. Dr. S. Richert Institute of Physical Chemistry University of Freiburg Albertstraße 21, 79104 Freiburg, Germany. [4]J. Zolg, Prof. Dr. A.J.C. Kuehne IQST – Center for Integrated Quantum Science and Technology Ulm University Albert-Einstein-Allee 11, 89081 Ulm, Germany. [5]These authors contributed equally: Larissa Schöneburg, Markus Gross. ✉e-mail: sabine.richert@uni-ulm.de; alexander.kuehne@uni-ulm.de

**Fig. 1 | Previously established stable organic radicals.** TTM, TTBrM, and **TTM-Cz** with $\phi$ = 88%, and the enantiomerically stable **TTBrM-Cz** with $\phi$ = 72%.

interesting molecular qubit[9]. To increase $\phi$, **TTM** has been coupled to electron donating substituents, disrupting the $C_3$ symmetry by producing a charge transfer (CT) excited state[10–13], strongly increasing its $\phi$ to 88 % (Fig. 1)[14,15]. Rendering this donor unit chiral produces light-emitting radicals with CPL, although with $g_{lum}$ only of order $10^{-4}$ [16]. Unfortunately, donor-functionalized **TTBrM** has not been synthesized to date, due to the reduced reactivity of bromine in radical-mediated nucleophilic aromatic substitution reactions, typically applied to attach the donor unit to a trityl unit. Moreover, the subsequent conversion to the radical is more difficult for **TTBrM** than for **TTM** because the required deprotonation of the central methine unit is not as effective with the more electropositive bromides. Accordingly, we have recently established that palladium-catalyzed cross-coupling can be employed to couple the radical, after selective iodine or bromine introduction to the *para*-site of **TTM**[17–19]. Realizing such donor functionalized **TTBrM** radicals would open up pathways to a new class of stable and highly luminescent trityl propellers. In contrast to previous reports with CPL in donor functionalized **TTM** radicals[16,20,21], the interesting CPL characteristics would not originate from the donor or a surrounding matrix but from the trityl propeller itself. Unfortunately, such donor-functionalized **TTBrM** radicals have not been synthesized, despite their interesting properties that have to date only been investigated theoretically[22–24].

Here, we demonstrate that carbazole (Cz)-functionalized **TTBrM** radicals can be obtained via Buchwald-Hartwig cross-coupling followed by the deprotonation-oxidation mechanisms. The resulting **TTBrM-Cz** radicals exhibit a strong $\phi$ of up to 72 % similar to **TTM-Cz**. Enantiomeric resolution yields stable radicals with CPL and $g_{lum}$ of up to $6.5 \times 10^{-4}$, opening up this new class of light-emitting radicals for further exploration.

## Results And Discussion

### Synthesis of donor functionalized TTBrM radicals

To generate the carbazole functionalized **TTBrM** compounds, we start from synthesizing the closed-shell **HTTBrM** radical precursor by Friedel-Crafts alkylation (the synthesis is described elsewhere)[7].

Notably, for the preparation of **HTTBrM**, an alternative protocol employing hexafluorobenzene as the reaction medium, affords markedly improved reproducibility and, owing to the reduced reaction temperature (80 °C versus 120 °C in the melt), minimizes competing side reactions (see SI). In contrast to the related **TTM-Cz** compound, Buchwald-Hartwig coupling between the **TTBrM** radical and the carbazole derivatives (**Cz, MeCz, Me₂Cz**) is not successful. Therefore, we perform Buchwald-Hartwig coupling between the Cz-derivatives and the **HTTBrM**-precursor and subsequently convert to the radical, via the established two-step deprotonation and oxidation sequence (Fig. 2)[16,25,26].

In optimizing this conversion, we find that sterically demanding bases (such as potassium *tert*-butoxide) are not suitable for deprotonation. By contrast, finely ground potassium hydroxide in combination with 18-crown-6 proves effective. The crown ether complexes the potassium ion and therefore activates the hydroxide ion, enabling deprotonation and formation of the **TTBrM**-anion under relatively mild conditions[27]. The oxidation to the radical is straight forward using *p*-chloranil (see Fig. 2 and the synthetic procedures in the SI). This approach represents a practical route to **TTBrM**-based radicals bearing electron-rich aryl substituents in high yields > 90 %, expanding the accessible scope of this class of **TTBrM** compounds.

### Optical, electronic, and spin properties of the TTBrM radicals

To gain deeper insight into the fundamental electronic and photophysical properties of the radicals, a combination of electron paramagnetic resonance (EPR) spectroscopy, optical, electrochemical, and computational studies are employed. Continuous wave (cw)-EPR spectroscopy reveals broad signals without hyperfine splitting, with $g$ values of 2.0063 to 2.0059, see Fig. 3a. The unsubstituted **TTBrM** radical (without donor moiety) shows a nearly identical value of $g$ = 2.0065. By contrast, **TTM** and **TTM-Cz** exhibit hyperfine splitting and show a similarly small shift in $g$ for the Cz donor attachment; however, at slightly lower values of 2.0035 and 2.0032, respectively (see the whole set of cw-EPR spectra in Fig. S1 in the SI)[8]. The small shift in the $g$-values upon Cz substitution indicates that the unpaired

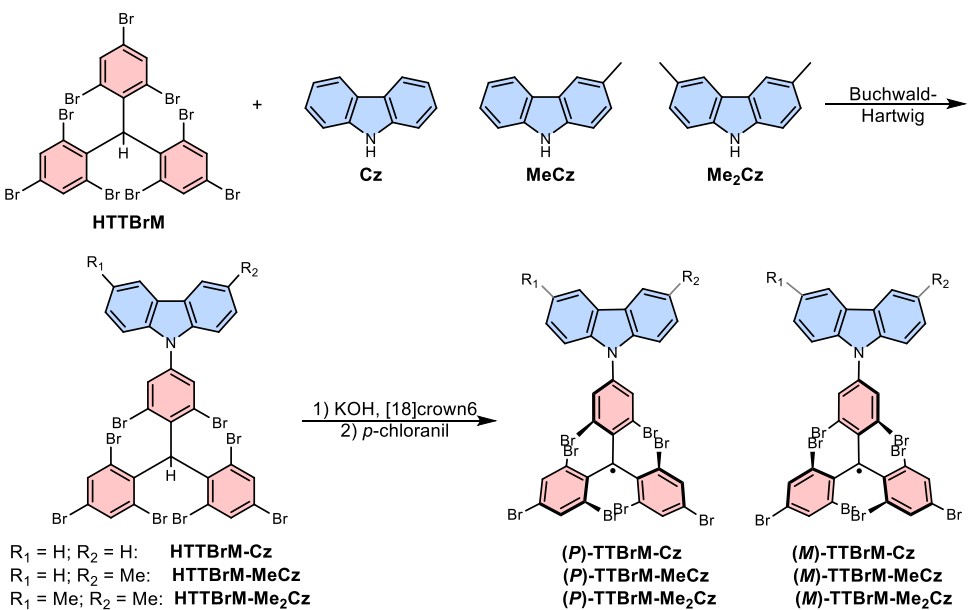

**Fig. 2 | Donor functionalized TTBrM radicals.** Pathway to synthesize carbazole functionalized **TTBrM** radicals via Buchwald-Hartwig coupling of Cz to the **HTTBrM** radical precursor, followed by deprotonation and oxidation. The 3,6-positions are successively methylated to adjust the donor strength of the carbazole moiety.

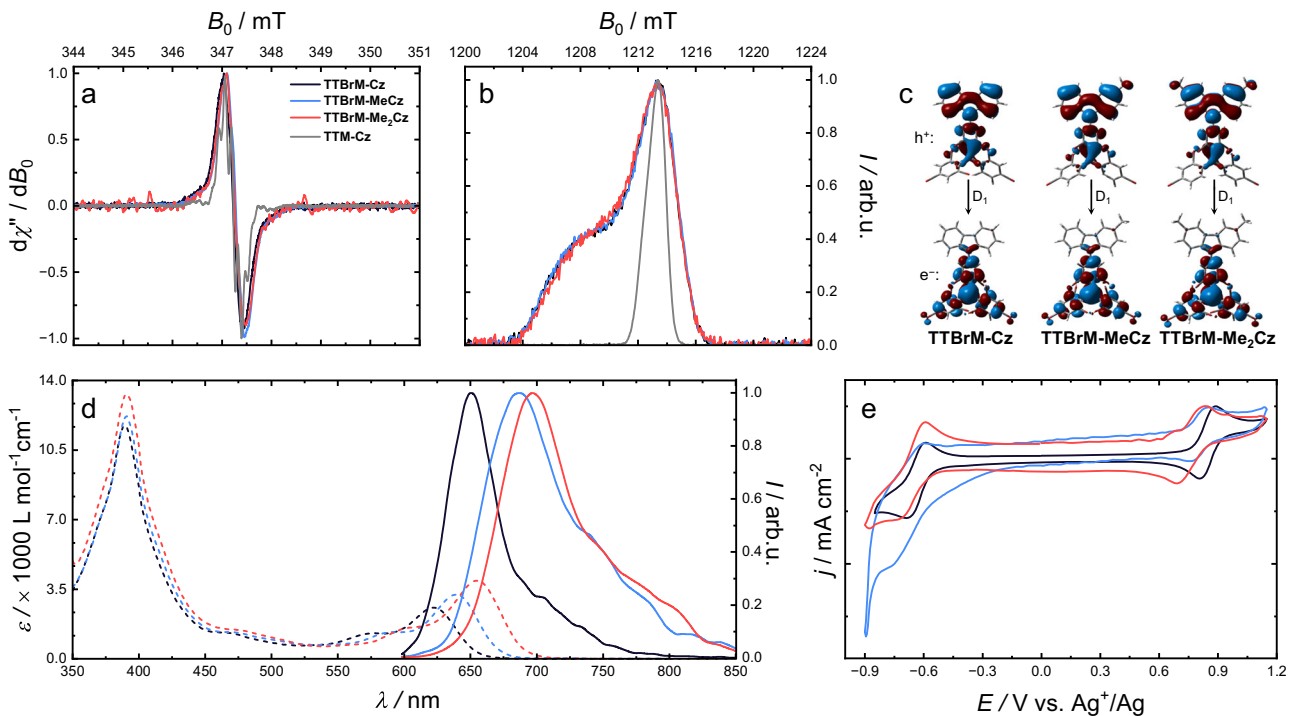

**Fig. 3 | Magnetic, computational, optical, and electrochemical data. a** cw-EPR signal recorded at the X-band (9.75 GHz) at 295 K of **TTBrM-Cz** (black), **TTBrM-MeCz** (blue), **TTBrM-Me$_2$Cz** (red), and **TTM-Cz** (grey) as a reference. **b** Electron spin echo (ESE) detected field-swept EPR spectra recorded at the Q-band (34 GHz) in frozen toluene solution at 80 K. **c** NTOs of the $D_1$ transition for the three donor-functionalized TTBrM radicals on the PBE1PBE/6-311 + + G** level of theory. **d** Absorption spectra in cyclohexane solutions (c - $10^{-5}$ mol L$^{-1}$) (dashed lines) and normalized photoluminescence spectra in cyclohexane solutions (c - $10^{-5}$ mol L$^{-1}$) (solid lines). **e** Normalized cyclic voltammograms of the radicals in methylene chloride at a scan rate of 10 mV s$^{-1}$ with [Bu$_4$N]$^+$[PF$_6$]$^-$ (0.1 M) as electrolyte.

electron remains localized on the trityl (**TTM/TTBrM**) core, with negligible delocalization into the donor unit. By contrast, the significant increase in $g$-value upon halogen substitution (Cl → Br) can be attributed to enhanced spin-orbit coupling (SOC) introduced by the heavier bromine atoms (cf. Fig. S2 in the SI). These findings confirm that the $g$-value is primarily governed by local SOC effects at the radical center, while distant donor groups such as Cz have only minimal influence unless directly involved in spin delocalization[28].

Previous studies have shown that the **TTBrM** radical exhibits significantly extended phase memory times $T_m$ compared to **TTM**, rendering **TTBrM** interesting as a molecular qubit[9]. To determine $T_m$ and the spin-lattice relaxation time $T_1$, we perform pulse EPR measurements in frozen toluene solution at 80 K (see Fig. 3b and Figs. S2–S4 in the SI). We are able to reproduce the reported $T_1$ for **TTBrM**, obtaining a value of 0.05 ms (literature: $T_1$ = 0.07 ms) (Fig. S3). However, the measured phase memory time $T_m$ = 1.5 ± 0.1 µs is

significantly shorter than the reported literature value ($T_m = 17.0$ μs) (Fig S4)[9]. For comparison, the corresponding relaxation times that we determine for **TTM** ($T_1 = 0.80$ ms; $T_m = 4.0 \pm 0.1$ μs) and **TTM-Cz** ($T_1 = 1.05$ ms; $T_m = 4.1 \pm 0.1$ μs) are markedly longer than those of **TTBrM**. The overall reduction in spin coherence is not unexpected for brominated radicals relative to their chlorinated analogues. This is attributed to bromine's higher atomic mass, which enhances SOC and thus increases spin relaxation rates, leading to shorter phase memory times. The discrepancy between our measured and the reported $T_m$ values for **TTBrM** may arise from solvent effects, variations in temperature and measurement conditions, or from differences in sample purity.

When we functionalize the **TTBrM** with our three different Cz-based electron donors, we observe slightly increased $T_1$ values compared to the unsubstituted **TTBrM** with $T_1 = 0.07$ ms, 0.09 ms, and 0.10 ms, respectively, for **TTBrM-Cz, TTBrM-MeCz**, and **TTBrM-Me$_2$Cz** (Table 1 and Figure S3). No significant variation in $T_m$ is observed with Cz functionalization and with successive methylation of the Cz-donor, with $T_m$ values ranging between $1.5 \pm 0.1$ μs and $1.6 \pm 0.1$ μs (see Fig. S4). These coherence times can be slightly increased when measuring in deuterated toluene, with $T_m \sim 2.1$ μs, while $T_m$ of the **TTM-Cz** reference is increased to $4.7 \pm 0.1$ μs (Fig. S5 in the SI). The results demonstrate that methylation of the carbazole does not negatively impact the phase memory times.

To further investigate the effect of methylation of the Cz-unit on the optical properties, we turn to UV-vis absorption spectroscopy of our compounds in cyclohexane solution (Fig. 3d). The absorption spectra are typical for donor functionalized trityl radicals, with a strong absorption band in the near UV spectrum, here at $\lambda_{abs} \sim 390$ nm ($\lambda_{abs} \sim 375$ nm for **TTM-Cz**)[14,29]. Moreover, we also observe the typical weak absorption feature in the red spectral region, at $\lambda_{abs} = 623$ nm for **TTBrM-Cz**, 640 nm for **TTBrM-MeCz**, and 655 nm for **TTBrM-Me$_2$Cz** (Fig. 3d and Table 1). With increasing methylation, the donor strength of the Cz units will increase because of the inductive +$I$-effect of the methyl groups. It is therefore understandable that with increasing methylation, we are increasing the Cz-donor strength and $\lambda_{abs}$ is systematically red shifted. For successive methylation of the Cz-donor, the extinction coefficient $\varepsilon$ of this red band increases, with respect to the strong band in the near UV. The spectral shift and stronger absorption suggest an increasing charge-transfer (CT) character of the absorption and a decreasing energy gap between the highest doubly occupied molecular orbital (HDMO) and the singly occupied molecular orbital (SOMO) for the increasing donor strength (HDMO-SOMO gap). To better elucidate the photophysical properties, UV-vis-absorption and photoluminescence spectra are recorded in solvents of increasing polarity (cyclohexane → ethyl acetate). While the absorption spectra remain less affected by the increased polarity of the solvents, the photoluminescence spectra exhibit a clear and systematic bathochromic shift, consistent with a stabilized CT excited state. In strongly polar solvents, the emission intensity is significantly reduced, indicative of enhanced non-radiative decay (Figs. S12–S14 in the SI). Concomitantly, in the UV-vis spectra, the low-energy absorption band becomes broader and less structured with increasing polarity. These

observations provide direct experimental evidence for the CT character of the D$_1$ transition (cf. Fig. 3c).

To rationalize these observations, density functional theory (DFT) and time-dependent (TD-)DFT calculations are performed. The computed excitation energies ($E_{ex} \approx 1.9$–2.0 eV) and oscillator strengths ($f \approx 0.11$–0.13) slightly increase with rising donor strength (Table 2). The dihedral angles between the donor and acceptor units ($\gamma_{DA}$) of the DFT optimized ground state of the radicals are very similar across the series (**TTBrM-Cz**: $\gamma_{DA} = 68.4°$, **TTBrM-MeCz**: $\gamma_{DA} = 67.0°$, **TTBrM-Me$_2$Cz**: $\gamma_{DA} = 66.6°$), indicating that electronic rather than geometric effects dominate the observed trends (see Fig. 3c and Table 2). Thus, the photophysical behavior appears to be primarily driven by electronic tuning through the donor substitution, rather than changes in conjugation or steric orientation.

Accordingly, the computed natural transition orbitals (NTOs, Fig. 3c) reveal a growing spatial separation between hole and electron densities along the series, reflecting the increased CT character from **TTBrM-Cz** to **TTBrM-Me$_2$Cz**. The excellent agreement between simulated and experimental UV-vis spectra further supports the accuracy of the computational model and confirms the CT nature of the transitions (Fig. 3d and Figs. S9–S11 in the SI). We hypothesize that the CT state is stabilized by the stronger electron-donating ability of the methyl-substituted donor units, where the inductive +$I$-effect raises the energy of the HDMO (in the radical ground state), while localizing it predominantly on the donor. This is in agreement with our calculated ground-state frontier molecular orbitals of all three radicals, whereas the lowest unoccupied molecular orbital (LUMO) remains localized on the TTBrM unit (Figs. S6–S8 in the SI).

This observation is also consistent with the decreasing ionization energies ($IE$) determined specifically for the isolated Cz-moieties. For these donor fragments we compute the $IE$ via TD-DFT as the energy difference between their cationic and neutral forms. A decrease in $IE$ is consistent with increasing electron donating strength of the Cz-groups (see Table 2).

To further support our hypothesis, we perform cyclovoltammetry (CV) using a three-electrode-setup with Ag/Ag$^+$ as a reference and [Bu$_4$N]$^+$[PF$_6$]$^-$ (0.1 M) as the electrolyte in anhydrous methylene chloride (Fig. 3e and Fig. S15 in the SI). All **TTBrM-Cz** radicals exhibit two well-defined and electrochemically distinct redox processes. A reversible reduction at approximately $-0.7$ V that can be attributed to the **TTBrM** radical center. Naturally, this feature is absent in the cyclic voltammograms of the corresponding individual carbazoles. By contrast, the oxidation wave is assigned to the carbazole donor unit and shows a systematic cathodic shift upon increasing methyl substitution, progressing from $E_{ox} = +0.89$ V for **TTBrM-Cz** to $+0.85$ V for **TTBrM-MeCz** and $+0.83$ V for **TTBrM-Me$_2$Cz** (vs. Ag/Ag$^+$). This trend closely follows the donor strengths and is consistent with the oxidation potentials of the respective carbazoles (Fig. S15 in the SI), as well as with the computed ionization energies (Table 2).

### Chiroptical properties of the TTBrM radicals

Like **TTBrM**, also our Cz-donor functionalized **TTBrM** radicals can be resolved into their respective enantiomers using chiral high-

## Table 1 | Electronic and spin resonance characterization

| | $\lambda_{abs}$ / nm | $\lambda_{PL}$ / nm | $\varepsilon$ / L mol cm$^{-1}$ | $\phi$ / % | $g$ | $\tau$ / ns (in Ar) | $k_r$ / $10^6$ s$^{-1}$ | $k_{nr}$ / $10^6$ s$^{-1}$ | $t_{1/2}$ / min |
|---|---|---|---|---|---|---|---|---|---|
| TTBrM-Cz | 390;623 | 646 | 3009 | 72 | 2.0063 | $38.89 \pm 0.02$ | 18.5 | 7.2 | 2 |
| TTBrM-MeCz | 391;639 | 679 | 3839 | 70 | 2.0059 | $37.36 \pm 0.02$ | 18.8 | 8.0 | 15 |
| TTBrM-Me$_2$Cz | 391;655 | 688 | 4679 | 62 | 2.0060 | $32.57 \pm 0.02$ | 19.0 | 11.7 | 49 |

All optical measurements are performed in cyclohexane solutions (c ~ 10$^{-5}$ mol L$^{-1}$). The extinction coefficients for the D$_0$ → D$_1$ transition are determined from the mean of three individual measurements at defined concentrations. The $g$ values are obtained by cw-EPR. The PL lifetimes and decay rates $k_r$ and $k_{nr}$ are determined by TCSPC and the half-life $t_{1/2}$ by photostability measurements, measured in toluene, under UV-irradiation.

**Table 2 | Computational, electrochemical, and chiroptical parameters**

|  | Donor $IE$ / eV | $E_{ex}$ / eV | $E_{ox}$ / V (vs. Ag/Ag$^+$) | $f$ | $\gamma_{DA}$ / ° | $g_{abs}$ / 10$^{-4}$ | $g_{lum}$ / 10$^{-3}$ | $B_{CPL}$ |
|---|---|---|---|---|---|---|---|---|
| TTBrM-Cz | 7.5 | 2.0 | 0.89 | 0.11 | 68.4 | 2.7 | 0.53 | 0.57 |
| TTBrM-MeCz | 7.3 | 1.9 | 0.85 | 0.12 | 67.0 | 2.9 | 0.58 | 0.69 |
| TTBrM-Me$_2$Cz | 7.1 | 1.9 | 0.83 | 0.13 | 66.6 | 3.1 | 0.65 | 0.94 |

Ionization energies ($IE$) for the donor moieties are obtained by DFT as the difference between the cationic and the neutral form. The excitation energies ($E_{ex}$) and oscillator strengths ($f$) are computed using TD-DFT. Oxidation potentials ($E_{ox}$) are determined by cyclic voltammetry vs. Ag/Ag$^+$. The dihedral angles between the donor and acceptor units ($\gamma_{DA}$) are determined in Avogadro using DFT-optimized geometries. The dissymmetry factors ($g_{abs}$, $g_{lum}$) and brightness of circularly polarized luminescence ($B_{CPL}$) are obtained from CD and CPL measurements, maximum values are shown here.

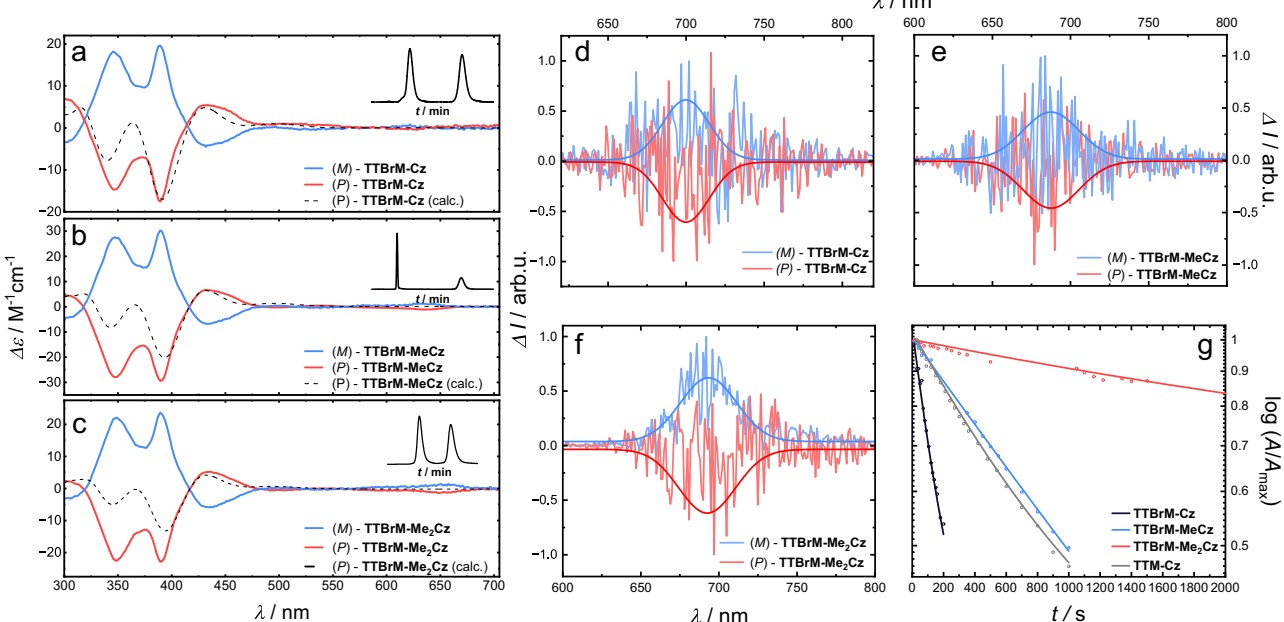

**Fig. 4 | Chiroptical data and photostability measurements. a–c** Circular dichroism spectra of **a TTBrM-Cz**, **b TTBrM-MeCz**, and **c TTBrM-Me$_2$Cz** in cyclohexane. For assigning of the obtained spectra to the respective (*M*)- and (*P*)-enantiomers, we calculated the CD spectrum of the (*P*)-enantiomer using TD-DFT. The calculated spectra are scaled along the y-axis to match the experimental values. **d–f** Circularly polarized luminescence as $\Delta I$ of **d TTBrM-Cz** in toluene, **e TTBrM-MeCz**, and **f TTBrM-Me$_2$Cz** both in cyclohexane with additional Gauss Fit as a guide to the eye. **g** Photostability measurements in degassed toluene under UV irradiation (400 nm at 70 mW/cm$^2$) with **TTBrM-Cz** (black), **TTBrM-MeCz** (blue), **TTBrM-Me$_2$Cz** (red), and **TTM-Cz** (grey) as a reference.

performance liquid chromatography (HPLC) (see details in the SI, Table S3 and Figs. S17–S19). We obtain excellent separation between the HPLC peaks on a chiral Daicel Chiralpak® IB N-3 column, and we receive the resolved enantiomers (see HPLC trace as insets in Fig. 4a–c). In contrast to **TTM** derivatives, our Cz-functionalized **TTBrM** radicals are stable against racemization at room temperature and the enantiomers exhibit circular dichroism (CD) with nicely mirror-imaged spectra and a maximum dissymmetry factor of $g_{abs} = 3.1 \times 10^{-4}$ (Table 2 and Fig. 4a-c). TD-DFT systematically underestimates $g_{abs}$, within the typical error margins of the method (see Table S2 in the SI). We can clearly observe the Cotton effect in the CD spectra at $\lambda_{abs} = 318$ nm and 416 nm, originating from the chirality of the **TTBrM** radical propellers. By comparing the theoretical CD spectrum of the (*P*)-enantiomers (obtained by TD-DFT) with the experimental data, the blue curves in Fig. 4 can be assigned to the (*M*)-enantiomer, and the red curves to the (*P*)-enantiomer (Fig. 4a-c). We test the thermal stability of our enantiomers by heating the sample in 10 °C steps and recording the CD spectrum. All three radicals, **TTBrM-Cz, TTBrM-MeCz**, and **TTBrM-Me$_2$Cz**, are stable up to a temperature of $T_{rac} > 80$ °C. After four hours at this temperature, only 5% of the pure enantiomer has racemized (Fig. S20 in the SI).

The effect of the donor strength variation by methylation is also observed in the photoluminescence (PL) spectra of the prepared cyclohexane solutions of our donor functionalized **TTBrM** radicals. **TTBrM-Cz** exhibits a photoluminescence maximum at $\lambda_{PL} = 646$ nm, **TTBrM-MeCz** at $\lambda_{PL} = 679$ nm, and **TTBrM-Me$_2$Cz** at $\lambda_{PL} = 688$ nm (Fig. 3d). Compared to **TTM-Cz** with $\lambda_{PL} = 628$ nm the brominated derivatives exhibit substantially red-shifted emission, consistent with the increased CT character and the reduced optical gaps. Despite these shifts, all compounds display high $\phi$, ranging from 62 % for **TTBrM-Me$_2$Cz** to 70 % for **TTBrM-MeCz** to 72 % for **TTBrM-Cz**, indicative of efficient radiative decay.

With such high $\phi$, we also investigate the circularly polarized photoluminescence (CPL) of our chiral radical propellers. Surprisingly, we are not able to determine CPL in cyclohexane solutions, despite having recorded clear CD spectra in this non-polar solvent. We turn to toluene solutions, in which we can record mirror-imaged CPL spectra for the resolved enantiomers with $g_{lum}$ dissymmetry factors of order 10$^{-4}$ (Table 2 and Fig. 4d–f). These values are in excellent agreement with $g_{lum}$ values determined via TD-DFT after geometrical optimization of our donor functionalized TTBrM radicals in the excited state (Table S2).

To explain the lower $\phi$ for the methylated radicals, we hypothesize that methylation could open up non-radiative relaxation channels. To elucidate these relaxation processes, photoluminescence lifetime measurements ($\tau$) are performed to extract the radiative ($k_r$) and non-

radiative ($k_{nr}$) rate constants (Figs. S21–S23 and Table 1). Whereas the $k_r$ is almost constant for the three radicals, $k_{nr}$ clearly increases for successively methylated radicals (**TTBrM-MeCz, TTBrM-Me$_2$Cz**), suggesting that non-radiative decay pathways are most suppressed in **TTBrM-Cz,** leading to the enhanced $\phi$.

Interestingly, photostability testing, where we irradiate degassed cyclohexane solutions of our radicals with a 5 W UV-lamp ($\lambda_{PL} = 400$ nm, $E = 70$ mW / cm$^2$), reveals a dramatic increase in the stability of the **TTBrM** radical with methylation of the Cz-donor. In cyclohexane solution, the donor-functionalized **TTBrM** radicals exhibit photostability half-lives ($t_{1/2}$) increasing from about 1 s for **TTBrM-Cz** to $t_{1/2} = 3$ s for **TTBrM-MeCz** to $t_{1/2} = 24$ s for **TTBrM-Me$_2$Cz** (Fig. 4c and Figs. S24–S26 in the SI). The higher photostability of the methylated donors may arise from their larger $k_{nr}$, and shorter excited-state lifetimes. This larger $k_{nr}$ could reduce the formation of reactive intermediates under photoexcitation. These short photostability half-lives may also be responsible for the insufficient signal during the relatively long CPL measurements. By contrast, **TTM-Cz** shows an intermediate stability half-life of $t_{1/2} = 4$ s, highlighting the stabilizing effect of the shielding methyl groups on the Cz-donor functionalized **TTBrM** radical. Changing the solvent to degassed toluene dramatically increases the photostability, yielding $t_{1/2} = 123$ s for **TTBrM-Cz**, 902 s for **TTBrM-MeCz**, 2593 s for **TTBrM-Me$_2$Cz** and 558 s for **TTM-Cz**, indicating that the higher polarity of toluene stabilizes the excited state of the radicals.

In summary, we have introduced highly luminescent **TTBrM**-radicals with stable chirality and good photostability. During the submission of this manuscript, we became aware of another study reporting the synthesis of a **TTBrM-Cz** radical[30]. By overcoming the intrinsic low emissivity of **TTBrM**, while preserving spin coherence and axial chirality, our work removes a key obstacle toward optical spin readout in **TTBrM**-based molecular qubits. As such, these donor functionalized **TTBrM** radicals represent interesting building blocks for spintronic applications and for molecular qubits.

## Methods

Detailed description of the methods used, synthetic procedures, and details on the computational study are provided within Supplementary Information. The authors have cited additional references within the Supplementary Information[31–33].

### Chemicals and solvents

All chemicals were used as received without further purification. Sodium tert-butoxide (NaOtBu) was obtained from abcr. 9H-Carbazole (Cz), Pd$_2$(dba)$_3$, anhydrous chloroform (CHCl$_3$), hexafluorobenzene, and potassium hydroxide (KOH) were purchased from Aldrich. 1,3,5-Tribromobenzene, 18-crown-6, and p-chloranil were supplied by Apollo Scientific. (tBu)$_3$PH·BF$_4$, 3-methyl-9H-carbazole (MeCz), and 3,6-dimethyl-9H-carbazole (Me2Cz) were obtained from BLD. Aluminum bromide was purchased from TCI. All reactions are performed under an inert atmosphere.

### Flash column chromatography

Purification via column chromatography is carried out with Silica 60 (particle size: 0.04-0.063mm) provided by Macherey-Nagel; the solvents were distilled beforehand. Aside from that, a Puriflash Interchim 450 with a silica column F0025, particle size 30 μm is used.

### High resolution mass spectrometry (HRMS)

Atmospheric pressure chemical ionization (APCI), laser desorption/ionization (LDI), and matrix-assisted laser desorption/ionization (MALDI) high-resolution mass spectrometry (HRMS) is performed on a Bruker solariX Fourier transform ion cyclotron resonance (FTICR) mass spectrometer. For MALDI, trans-2-[3-(4-tert-butyl-phenyl)-2-methyl-2-propenylidene]malononitrile (DCTB) is used as a matrix.

### Nuclear magnetic resonance (NMR) spectroscopy

NMR spectra are obtained by using the Bruker AVANCE NEO 400 or AVANCE NEO 600, and CD$_2$Cl$_2$ is chosen as the solvent.

### UV/vis absorption spectroscopy

UV-Vis spectra are measured with a Perkin Elmer Lambda 365.

### Fluorescence spectroscopy

Photoluminescence spectra are obtained by Perkin Elmer FL6500.

### Photoluminescence quantum yields

The quantum yield is determined on a HAMAMATSU Quantaurus-QY. All optical data are measured in cyclohexane (spectrophotometic grade) with a concentration of ~$10^{-5}$ M.

### Cyclic voltammetry

For the cyclovoltammetry (CV) measurements, a potentiostat from Ivium Technologies (IviumStat.h) was used. A platinum disc was used as the working electrode, a platinum wire as the counter electrode, and a non-aqueous Ag/Ag$^+$ electrode as the reference.

### Chiral stationary phase HPLC

The AGILENT analytical series with a chiral stationary phase of Daicel CHIRALPAK® IB N-3 (250 ×4.6 mm, particle size 3 μm) and Phenomenex Lux® i-Amylose-3 (250 ×4.6 mm, particle size 5 μm) column at a flow rate of 0.5–1 ml/min and a concentration of 1 mg/ml is used to separate the enantiomers.

### Electronic circular dichroism and circularly polarized luminescence spectroscopy

The Olis DSM 245 spectrophotometry system is used for measuring chiroptical properties. For ECD measurements a fixed slit width of 5 mm and an integration time of 0.5 s were used. CPL spectra were recorded with slits of 3.16 mm, an excitation wavelength of 390 nm and an integration time of 2 s. All solutions were deoxygenated prior to measurement.

### DFT

The theoretical calculations were performed with Gaussian 16 (Revision C.01)[33]. The geometry optimization was performed on the PBE1PBE/6-311 + + G** level of theory. To account for non-covalent interactions, the empirical dispersion correction GD3BJ is applied. Furthermore, the solvent environment is simulated using the SMD continuum model with cyclohexane as the solvent (SCRF = SMD, Solvent = Cyclohexane). TD-DFT is performed on the same level of theory and under the same conditions, for N = 50 states. Excited state calculations were subsequently carried out at the same level of theory and geometry optimization of the excited state was performed using the TD-DFT results as a starting point with an increased SCF convergence limit (scf = (maxcycle = 3000)). The number of computed states was set to five excited states with the first excited state as the reference state (TD = (NStates = 5, Root = 1)).

## Data availability

All data are available from the corresponding author upon request. Data generated or analyzed for this study are included in this manuscript and Supplementary Information. Data generated from DFT calculations have been deposited in the Zenodo database under accession code: https://doi.org/10.5281/zenodo.18268671.

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

## Acknowledgements

We thank Markus Lamla for recording the high-resolution mass spectra and Levin Göttinger for his project work on HTTBrM derivatives.

## Author contributions

L.S. and M.G. carried out the synthesis and characterization of the compounds. A.J.C.K. and S.R. conceptualized the project. M.E.A. supervised the synthesis. S.R. and P.T. conducted EPR experiments and analysis. J.Z. and L.S. performed (TD) DFT calculations. P.A.S. performed cyclic voltammetry, and B.P. performed fluorescence lifetime measurements. S.R. and A.J.C.K. provided supervision, infrastructure, and secured funding. L.S., M.G. and A.J.C.K. wrote the first draft and all authors revised and edited the manuscript.

## Funding

J. Z. acknowledges IQST for a PhD program within the IQST Graduate School @QuantumBW supported by the Baden-Württemberg Ministry of Science, Research, and Arts. A.J.C.K. discloses financial support from the Baden-Württemberg Stiftung gGmbH [ORQOSS project]. This work was supported by the Deutsche Forschungsgemeinschaft (DFG, German Research Foundation) [grant numbers 500226157, 445471845, 445471097 (A.J.C.K), 417643975, 536668010 (S.R.) and 577236137 (A.J.C.K. and S.R.)]. The authors acknowledge support by the state of Baden-Württemberg through bwHPC and the DFG through grant no INST 40/575-1 FUGG (JUSTUS 2 cluster). Open Access funding enabled and organized by Projekt DEAL.

## Competing interests

The authors declare no competing interests.
