## [Transparent Peer Review file · Nature Communications]

Highly Luminescent Carbazole-functionalized Tris(tribromophenyl)methyl Radicals with Stable Circularly Polarized Photoluminescence

Corresponding Author: Professor Alexander Kühne

Version 0:

Reviewer comments:

Reviewer #1

(Remarks to the Author)

In this work, Schöneburg et al. report three luminescent radicals based on a tris(2,4,6-tribromophenyl)methyl (TTBrM) skeleton. Unlike previous systems, the observed circular dichroism (CD) and circularly polarized luminescence (CPL) originate from the inherently chiral radical core rather than from the donor moieties, thereby creating greater scope for enhancing fluorescence efficiency. This rational design results in a high fluorescence efficiency of up to 72% along with stable chiroptical properties. The manuscript is therefore suitable for publication, pending the following major revisions.

- 1) To provide additional DFT data (such as ground state orbital distributions and energy levels) in the Supporting Information.
- 2) To allow for a more complete comparison, cyclic voltammetry (CV) curves of the donor segments (Cz, MeCz, and Me₂Cz) should be measured and reported.
- 3) To better elucidate the photophysical properties, the absorption and photoluminescence spectra of the three radicals should be provided in different solvents, including toluene. Furthermore, the fluorescence lifetime decays (Figures S14–S16) should be replotted with a logarithmic y-axis.
- 4) It is notable that no CPL signal was detected in hexane solution despite the high fluorescence efficiency. Can authors give a possible reason for this observation? Additionally, given that the reported dissymmetry factors (g values) remain at a level of approximately 10^{-4} , it is recommended to calculate the transition electric and magnetic dipole moments, as well as the corresponding simulated g values, to help identify the factors limiting CPL intensity.

Reviewer #2

(Remarks to the Author)

In the present manuscript, entitled “Highly Luminescent Carbazole-functionalized Tris(tribromophenyl)methyl Radicals with Stable Circularly Polarized Photoluminescence”, Prof. Alexander J. C. Kuehne and co-workers described the synthesis of three carbazole-functionalized tris(2,4,6-tribromophenyl)methyl (TTBrM) radicals and their characterization. In particular, the three compounds were synthesized by Buchwald-Hartwig reaction of HTTBrM with three different carbazole derivatives (Cz, MeCz, Me₂Cz), followed by deprotonation with KOH + [18]crown-6 and subsequent oxidation with p-chloranil. The three final compounds, named TTBrM-Cz, TTBrM-MeCz and TTBrM-Me₂Cz, were then subjected to a detailed characterization, involving EPR, optical, electrochemical and computational studies. The three radicals were then resolved into their respective enantiomers using chiral HPLC, and these were found to be stable against racemization for $T < 80$ °C. Finally, the compounds were subjected to chiroptical investigation, both in absorption (ECD) and emission (CPL).

The present work is definitely interesting for scientists working in the context of chiral organic materials, and the manuscript is clear and well written. However, these requirements are not sufficient to justify publication in a very high-impact journal such as Nature Communications. In this manuscript, the justification of the authors lies mainly in the fact that donor-functionalized tris(tribromophenyl)methyl radicals has not been synthesized to date, and that their work has allowed for the first time to obtain tris(tribromophenyl)methyl radicals, with propeller-type chirality, that are stable against racemization at room temperature.

Actually, a very recent work of Ken Albrecht and collaborators (a preprint published on Chemrxiv on 28 January 2026, DOI: 10.26434/chemrxiv.10001675/v1) reported the development of carbazole-functionalized tris(2,4,6-tribromophenyl)methyl

(TTBrM) radicals. In particular, one of the three compounds synthesized by Albrecht, called CzTTBrM, corresponds exactly to the compound TTBrM-Cz presented and studied in this work. I want to emphasize that I am absolutely convinced that the two research groups worked in parallel and independently: in fact, in the pre-print of Albrecht et al and in this work two synthetic procedures were used; moreover, the other two compounds in this work (TTBrM-MeCz and TTBrM-Me2Cz) are different from those synthesized by Albrecht (2CzTTBrM and 3CzTTBrM). This is truly an unfortunate coincidence! However, most of the characterizations conducted on TTBrM-Cz have already been reported in Albrecht's preprint for CzTTBrM, as well as the conclusions of the work related to stable CPL for tris(2,4,6-tribromophenyl)methyl radicals. It is quite clear, in light of this preprint by Albrecht and collaborators, that the main innovation and impact of this work, which would justify publication in Nature Communications, is missing. Therefore, I suggest transferring the present manuscript to a sectorial/more specialized journal, which I am sure will be happy to accept this work after proper peer-review.

Reviewer #3

(Remarks to the Author)
Dear Editor,

In this paper, Richert, Kuehne, et al. report on the functionalization of the well-known tris(tribromophenyl)methyl (TTBrM) radical with carbazole units, demonstrating the feasibility of the Buchwald–Hartwig cross-coupling on the parent brominated HTTBrM precursor, on a very interesting organic chemistry synthesis development. With a view to assessing the relevance of the resulting TTBrM-Cz radicals as potential spin-based qubits, the spin dynamics have been studied in frozen solutions via standard Hahn-echo and inversion recovery pulse sequences. The performance in terms of phase quantum coherence is one order of magnitude poorer than reported for the parent TTBrM radical (J. Mater. Chem. C, 2024, 12, 5150–5156). The absorption properties of the TTBrM-Cz radicals were examined and the trends of the changes to the minor bands in the visible region across the series are rationalized in terms of the donor properties of the Cz substituent. DFT calculations help in rationalizing this behavior. These conclusions are supported by the results from cyclic voltammetry. For all the TTBrM-Cz derivatives, the pure enantiomers can be separated by chiral HPLC, and as expected, the optically active species remain stable at room temperature against racemization. Accordingly, CD spectra can be obtained. In addition, the compounds exhibit photoluminescence (PL) with high quantum yields. On the other hand, circularly polarized PL is extremely poor. Photostability studies for the radicals have been performed in hexane and the $t_{1/2}$ times have been benchmarked as a function of the number of Me substituents on the carbazole (Cz) unit, with a direct correlation. A considerable increase of $t_{1/2}$ values is observed in toluene. Realizing the functionalization of the HTTBrM closed-shell precursor with carbazoles is interesting, considering that the Cl counterparts have been obtained previously in several instances via cross-coupling reactions (e.g.; J. Mater. Chem. B, 2024, 12, 6840–6846; Mater. Chem. Front., 2017, 1, 2132-2135; Mater. Horiz., 2019, 6, 1265-1270, none of them cited in the submitted draft) leading to interesting optical properties, including enhanced luminescence efficiency. Thus, in the analogous manner as for other aryl-bromides of high complexity (e.g. ACS Omega, 2024, 9, 50446–50457; J. Org. Chem. 2021, 86, 17594–17605), the Buchwald–Hartwig cross-coupling has proven effective for this functionalization. However, it must be emphasized that this was indeed hard to predict, considering the importance of the steric hinderence posed by the Br substituents to the oxidative insertion of Pd(0) into the C-Br bond. The resulting TTBrM-Cz exhibit moderate to poor phase memory times, as revealed from routine fits of the decay of Hahn-echo detected signals, ascribed precisely to the Br substituents and their effect on the SOC. The optical properties of the new Cz derivatives outperform these from the not functionalized radical precursors, as one could have expected from previous work on the Cl analogues (see for example J. Mater. Chem. B, 2024, 12, 6840–6846). Therefore, the potential improvements reported here with the new TTBrM-Cz derivatives do not justify publication in Nature Communications. The fact that the racemization in solution is slower with the Br versions than for the Cl ones is hardly relevant, since any implementation of spin-based qubits will take place on solid state hybrid architectures. Indeed, optical spin readout requires fulfilment of harsh conditions, such as the demonstration that the optical transitions couple differently with the spin states encoding a qubit or the ability to polarize the spin upon optical excitation. None of this is addressed in the current paper, which could be considered a real breakthrough.

I suggest addressing some minor issues before submitting the paper to a more specialized journal:

1. I could not find any indication of the concentration used for the EPR measurements, especially for the relaxation time measurements. It would be relevant to know whether the same concentration was used throughout, and whether it could be considered sufficiently low to neglect intermolecular dipole–dipole contributions to relaxation. Judging from the signal-to-noise ratio, the samples were likely quite dilute, but this should at least be explicitly stated.
2. The variation in phase memory time among the various Cz derivatives is very small (from ~1.5 μ s to ~1.7 μ s). The trend observed could easily change because of many possible sources of experimental error, for example, differences in glass formation of the toluene matrix, which are inherently difficult to reproduce perfectly. Since the observed trend is surprising and counter-intuitive, in-depth study and confidence intervals are necessary before any hypotheses are put forward. Certainly, the explanation given; “screening of spins from the decohering environment”, seems highly questionable. Methyl groups themselves are a source of relaxation, so it is difficult to imagine that a frozen toluene matrix contributes more to relaxation than the methyl groups of the molecule. This could, however, be tested straightforwardly by using deuterated toluene, where the contribution of solvent nuclear spin diffusion should be significantly reduced.
3. The CPL performance is very poor and heavily affected by very low signal-to-noise ratio. In fact, the Gauss fits as guide to the eye in figures d, e and f (especially the first two) are rather unconvincing and put into question the quantitative glum provided.

Reviewer #4

(Remarks to the Author)

Version 1:

Reviewer comments:

Reviewer #1

(Remarks to the Author)

The authors have properly addressed the proposed issues, and the manuscript is recommended for acceptance.

Reviewer #3

(Remarks to the Author)

Dear Editor,

After reading the revisions and explanations given by the authors in their letter, I recognize that they have made important efforts to address the referee's concerns. However, the paper brings essentially the same message as the initial one. I have a very similar opinion as that of Referee #2. The interesting organic synthesis development does not justify publication in Nature Communications. As mentioned by the authors, the enhanced photophysical properties of the Br derivatives were already established on the Cl analogues, while the spin dynamics are not really better. The improvement on the stabilization of the optical activity is expected, as has been observed before (Chem. - A Eur. J. 2020, 26, 3776–3781). In my opinion, this interesting paper, in its form after the revisions, should be transferred to a more specialized journal.

Reviewer #4

(Remarks to the Author)

REVIEWER REPORT(S):

Recommendation: Major revisions

Referee: 1

Comments:

In this work, Schöneburg et al. report three luminescent radicals based on a tris(2,4,6-tribromophenyl)methyl (TBrM) skeleton. Unlike previous systems, the observed circular dichroism (CD) and circularly polarized luminescence (CPL) originate from the inherently chiral radical core rather than from the donor moieties, thereby creating greater scope for enhancing fluorescence efficiency. This rational design results in a high fluorescence efficiency of up to 72% along with stable chiroptical properties.

The manuscript is therefore suitable for publication, pending the following major revisions.

Major points:

1. To provide additional DFT data (such as ground state orbital distributions and energy levels) in the Supporting Information.

Thank you for your valuable comment. We agree that these additional DFT data are useful. We now provide energy diagrams that also include ground state orbital distributions.

Changes to the manuscript (page 7):

*“...The excellent agreement between simulated and experimental UV-vis spectra further supports the accuracy of the computational model and confirms the CT nature of the transitions (see **Figures 3d** and **Figures S9-S11** in the SI). We hypothesize that the CT state is stabilized by the stronger electron-donating ability of the methyl-substituted donor units, where the inductive +I-effect raises the energy of the HOMO (in the radical ground state), while localizing it **predominantly** on the donor. **This is in agreement with our calculated ground-state frontier molecular orbitals of all three radicals, whereas the lowest unoccupied molecular orbital (LUMO) remains localized on the TBrM unit (see **Figures S6-S8** in the SI).***

This observation is also consistent with the decreasing ionization energies (IE) determined specifically for the isolated Cz-moieties. ...”

Changes to the Supporting Information (Figures S6-S8):

Figure S1: Ground state orbital distributions and corresponding energy levels of TTBm-Cz.

Figure S2: Ground state orbital distributions and corresponding energy levels of TTBm-MeCz.

Figure S3: Ground state orbital distributions and corresponding energy levels of **TTBrM-Me₂Cz**.

- To allow for a more complete comparison, cyclic voltammetry (CV) curves of the donor segments (Cz, MeCz, and Me₂Cz) should be measured and reported.

We thank the reviewer for this suggestion. We now provide cyclic voltammograms of the donors alongside the cyclic voltammograms for the donor functionalized radicals.

Changes to the manuscript (page 7-8):

“... To further support our hypothesis, we perform cyclovoltammetry (CV) using a three-electrode-setup with Ag/Ag⁺ as a reference and [Bu₄N]⁺[PF₆]⁻ (0.1 M) as the electrolyte in anhydrous methylene chloride (see **Figure 3e** and **Figure S15** in the SI). All **TTBrM-Cz** radicals exhibit two well-defined and electrochemically distinct redox processes. A reversible reduction at approximately -0.7 V that can be attributed to the **TTBrM** radical center. Naturally, this feature is absent in the cyclic voltammograms of the corresponding individual carbazoles. By contrast, the oxidation wave is assigned to the carbazole donor unit and shows a systematic cathodic shift upon increasing methyl substitution, progressing from $E_{ox} = +0.89$ V for **TTBrM-Cz** to +0.85 V for **TTBrM-MeCz** and +0.83 V for **TTBrM-Me₂Cz** (vs. Ag/Ag⁺). This trend closely follows the donor strengths and is consistent with the oxidation potentials of the respective carbazoles (see **Figure S15** in the SI), as well as with the computed ionization energies (see **Table 2**). ...”

Changes to the ESI (Figure S15):

Figure S15: CV measurements in methylene chloride of **TTBrM-Cz** (black), **TTBrM-MeCz** (blue), and **TTBrM-Me₂Cz** (red) in a three-electrode-setup with an Ag/Ag⁺ reference electrode and [Bu₄N]⁺[PF₆]⁻ (0.1 M) as electrolyte. The insets display the cyclic voltammograms of the corresponding donors.

3. To better elucidate the photophysical properties, the absorption and photoluminescence spectra of the three radicals should be provided in different solvents, including toluene. Furthermore, the fluorescence lifetime decays (Figures S14–S16) should be replotted with a logarithmic y-axis.

We thank the reviewer for this comment. Regarding the photophysical properties, we are happy to provide these additional data. We provide absorption and photoluminescence spectra in cyclohexane, toluene, dichloromethane, chloroform, and ethyl acetate. The CT character is supported by the pronounced positive solvatochromism observed in the photoluminescence spectra. We have adjusted the manuscript accordingly to account for the additional information on the CT excited state.

Furthermore, we thank the reviewer for the suggestion on the fluorescence lifetime decays. The figures have been updated accordingly.

Changes to the manuscript (page 6):

“ ... The spectral shift and stronger absorption suggest an increasing charge-transfer (CT) character of the absorption and a decreasing energy gap between the highest doubly occupied molecular orbital (HDMO) and the singly occupied molecular orbital (SOMO) for the increasing donor strength (HDMO-

SOMO gap). To better elucidate the photophysical properties, UV-vis-absorption and photoluminescence spectra are recorded in solvents of increasing polarity (cyclohexane → ethyl acetate). While the absorption spectra remain less affected by the increased polarity of the solvents, the photoluminescence spectra exhibit a clear and systematic bathochromic shift, consistent with a stabilized CT excited state. In strongly polar solvents, the emission intensity is significantly reduced, indicative of enhanced non-radiative decay (see **Figures S12-S14** in the SI). Concomitantly, in the UV-vis spectra, the low-energy absorption band becomes broader and less structured with increasing polarity. These observations provide direct experimental evidence for the CT character of the D_1 transition (cf. **Figure 3c**). ...”

Changes to the ESI (**Figures S12-S14, S21-S23**):

Figure S12: Solvent-polarity-dependent absorption and emission spectra of **TTBrM-Cz**. The spectra have been normalized to enable better comparability of the emission maxima and spectral shapes.

Figure S13: Solvent-polarity-dependent absorption and emission spectra of **TTBrM-MeCz**. The spectra have been normalized to enable better comparability of the emission maxima and spectral shapes.

Figure S14: Solvent-polarity-dependent absorption and emission spectra of **TTBrM-Me₂Cz**. The spectra have been normalized to enable better comparability of the emission maxima and spectral shapes.

Figure S21: TCSPC measurement of a degassed **TTBrM-Cz** solution to obtain the PL lifetime.

Figure S22: TCSPC measurement of a degassed **TTBrM-Me₂Cz** solution to obtain the PL lifetime.

Figure S23: TCSPC measurement of a degassed **TBrM-Me₂Cz** solution to obtain the PL lifetime.

- It is notable that no CPL signal was detected in hexane solution despite the high fluorescence efficiency. Can authors give a possible reason for this observation? Additionally, given that the reported dissymmetry factors (*g* values) remain at a level of approximately 10^{-4} , it is recommended to calculate the transition electric and magnetic dipole moments, as well as the corresponding simulated *g* values, to help identify the factors limiting CPL intensity.

We thank the reviewer for this insightful comment. The absence of a detectable CPL signal in cyclohexane, despite high fluorescence efficiency, can be attributed to a combination of limited solubility and reduced photostability in this solvent.

First, the solubility of the TBrM-Cz radicals in cyclohexane is significantly lower than in toluene, preventing us from reaching the concentrations required for reliable CPL measurements and reproducible g_{lum} values. Notably, a recent preprint by Albrecht et al. (see also response to Reviewer 2) reports g_{abs} and g_{lum} values exclusively in toluene, while other photophysical measurements were conducted in cyclohexane. This suggests that similar limitations may also apply in their case.

Second, we observe a pronounced solvent dependence of photostability. For example, TBrM-Me₂Cz exhibits a very short degradation half-life in cyclohexane ($t_{1/2} \sim 5$ s), whereas in toluene the half-life increases to 558 s (Figures S24-S26). We attribute this to stabilization of the polar charge-transfer (CT) excited state in the more polar solvent. In cyclohexane, insufficient stabilization of the CT state likely accelerates photodegradation, which is particularly detrimental for CPL measurements that require signal accumulation over several minutes.

Taken together, these factors prevent reliable CPL detection in cyclohexane.

Following the reviewer's suggestion, we performed TD-DFT calculations to estimate transition electric and magnetic dipole moments and to derive corresponding g_{abs} and g_{lum} values. The calculated values agree well with the experiment within the typical TD-DFT accuracy (± 1 order of magnitude). These results have been included in the Supporting Information (see Table S2).

The only notable deviation is observed for g_{abs} of TBrM-MeCz, which could not be reproduced within expected accuracy. We attribute this to subtle symmetry breaking introduced by the single methyl substituent, leading to conformational relaxation in the ground state that is not fully captured by DFT and may result in an underestimated rotatory strength. Interestingly, optimization of the charge-transfer excited state yields a g_{lum} value in good agreement with the experiment.

Changes to the manuscript (pages 8):

“...the enantiomers exhibit circular dichroism (CD) with nicely mirror-imaged spectra and a maximum dissymmetry factor of $g_{\text{abs}} = 3.1 \times 10^{-4}$ (see **Table 2** and **Figure 4a**). TD-DFT systematically underestimates g_{abs} , within the typical error margins of the method (see **Table S2** in the SI). We can clearly observe the Cotton effect in the CD spectra at $\lambda_{\text{abs}} = 318 \text{ nm}$ and 416 nm ,...”

Changes to the manuscript (page 9):

“... We turn to toluene solutions, in which we can record mirror-imaged CPL spectra for the resolved enantiomers with g_{lum} dissymmetry factors of order 10^{-4} (see **Table 2** and **Figure 4d-f**). These values are in excellent agreement with g_{lum} values determined via TD-DFT after geometrical optimization of our donor functionalized TTBm radicals in the excited state (see **Table S2**). ...”

Changes to the manuscript (page 10):

“... In cyclohexane solution, the donor-functionalized TTBm radicals exhibit photostability half-lives ($t_{1/2}$) increasing from about 1 s for TTBm-Cz to $t_{1/2} = 3 \text{ s}$ for TTBm-MeCz to $t_{1/2} = 24 \text{ s}$ for TTBm-Me₂Cz (see **Figure 4c** and **Figures S24-S26** in the SI). The higher photostability of the methylated donors may arise from their larger k_{nr} , and shorter excited-state lifetimes. This larger k_{nr} could reduce the formation of reactive intermediates under photoexcitation. These short photostability half-lives may also be responsible for the insufficient signal during the relatively long CPL measurements. By contrast, TTM-Cz shows an intermediate stability...”

Changes to the ESI (Table S2):

Table S2 Summary of the calculated dissymmetry factors for absorption (g_{abs}) and emission (g_{lum}) of the (*P*)-enantiomer radicals. $|\mu_e|$ is the electric transition dipole moment, $|\mu_m|$ is the magnetic transition dipole moment, θ is the angle between the electric and magnetic transition dipole moment vectors.

(P)	Absorption (ground state geometry)				Photoluminescence (excited state geometry)			
	$ \mu_e / 10^{-20}$ esu·cm	$ \mu_m / 10^{-20}$ erg G ⁻¹	$\theta / ^\circ$	g_{abs} 10^{-5}	$ \mu_e / 10^{-20}$ esu·cm	$ \mu_m / 10^{-20}$ erg G ⁻¹	$\theta / ^\circ$	g_{lum} 10^{-4}
TTBrM-Cz	382	0.0034	2.59	3.01	350	0.0147	3.14	1.68
TTBrM-MeCz	402	0.0189	1.57	0.07	202	0.0269	2.79	4.97
TTBrM-Me₂Cz	424	0.0025	0.99	1.31	207	0.0252	3.14	4.87

Referee: 2

Comments:

In the present manuscript, entitled “Highly Luminescent Carbazole-functionalized Tris(tribromophenyl)methyl Radicals with Stable Circularly Polarized Photoluminescence”, Prof. Alexander J. C. Kuehne and co-workers described the synthesis of three carbazole-functionalized tris(2,4,6-tribromophenyl)methyl (TTBrM) radicals and their characterization. In particular, the three compounds were synthesized by Buchwald-Hartwig reaction of HTTBrM with three different carbazole derivatives (Cz, MeCz, Me₂Cz), followed by deprotonation with KOH + [18]crown-6 and subsequent oxidation with p-chloranil. The three final compounds, named TTBrM-Cz, TTBrM-MeCz and TTBrM-Me₂Cz, were then subjected to a detailed characterization, involving EPR, optical, electrochemical and computational studies. The three radical were then resolved into their respective enantiomers using chiral HPLC, and these were found to be stable against racemization for T < 80 °C. Finally, the compounds were subjected to chiroptical investigation, both in absorption (ECD) and emission (CPL). The present work is definitely interesting for scientists working in the context of chiral organic materials, and the manuscript is clear and well written. However, these requirements are not sufficient to justify publication in very a high-impact journal such as Nature Communications. In this manuscript, the justification of the authors lies mainly in the fact that donor-functionalized tris(tribromophenyl)methyl radicals has not been synthesized to date, and that their work has allowed for the first time to obtain tris(tribromophenyl)methyl radicals, with propeller-type chirality, that are stable against racemization at room temperature.

Actually, a very recent work of Ken Albrecht and collaborators (a preprint published on Chemrxiv on 28 January 2026, DOI: 10.26434/chemrxiv.10001675/v1) reported the development of carbazole-functionalized tris(2,4,6-tribromophenyl)methyl (TTBrM) radicals. In particular, one of the three compounds synthesized by Albrecht, called CzTTBrM, corresponds exactly to the compound TTBrM-Cz presented and studied in this work. I want to emphasize that I am absolutely convinced that the two research groups worked in parallel and independently: in fact, in the pre-print of Albrecht et al and in this work two synthetic procedures were used; moreover, the other two compounds in this work (TTBrM-MeCz and TTBrM-Me₂Cz) are different from those synthesized by Albrecht (2CzTTBrM and 3CzTTBrM). This is truly an unfortunate coincidence! However, most of the characterizations conducted on TTBrM-Cz have already been reported in Albrecht's preprint for CzTTBrM, as well as the conclusions of the work related to stable CPL for tris(2,4,6-tribromophenyl)methyl radicals. It is quite clear, in light of this preprint by Albrecht and collaborators, that the main innovation and impact of this work, which would justify publication in Nature Communications, is missing. Therefore, I suggest transferring the present manuscript to a sectorial/more specialized journal, which I am sure will be happy to accept this work after proper peer-review.

We thank the reviewer for the time taken to assess our manuscript and for highlighting the recent preprint by Ken Albrecht and co-workers. We fully agree that this situation reflects an independent and simultaneous development by two groups, which indeed underlines the timeliness and relevance of this emerging class of donor-functionalized TTBrM radicals.

While one compound (TTBrM-Cz) overlaps with CzTTBrM reported in the preprint, we would like to emphasize that our work provides a distinct and substantially extended contribution in several important aspects:

1. We introduce an efficient and general synthetic strategy to access donor-functionalized TTBrM radicals, achieving significantly improved overall yields (52–58% over two steps). This establishes a practical route that facilitates broader exploration of this materials class.
2. Our study includes two additional, structurally distinct derivatives (TTBrM-MeCz and TTBrM-Me₂Cz) that are not reported in the Albrecht *et al.* preprint. These compounds allow us to systematically investigate substituent effects on photophysical, spin, and chiroptical properties.
3. We provide a comprehensive characterization of spin properties and excited-state behavior, including detailed EPR analysis, kinetic studies, and structure-property relationships, which are not addressed in comparable depth in the preprint by Albrecht *et al.*

4. Our work delivers a systematic chiroptical investigation, including enantiomer separation, configurational stability, CD, and CPL measurements across a series of related compounds, enabling broader insight into the origin and control of CPL in TTB_rM-based radicals.

Taken together, we believe that our manuscript goes significantly beyond the scope of the preprint by Albrecht *et al.*, providing a general synthetic platform, a broader molecular library, and a more in-depth mechanistic and chiroptical understanding of these systems.

We note that both studies were conducted independently and disclosed in close temporal proximity (our preprint was deposited on ChemRxiv on January 27, 2026 (<https://doi.org/10.26434/chemrxiv.10001626/v1>) – one day before the deposition by Albrecht *et al.*), further supporting the view that this is an emerging and highly relevant research direction.

For these reasons, we respectfully believe that the novelty and impact of our work remain fully justified for publication.

We now refer to the preprint by Albrecht *et al.* in the Summary.

Changes to the manuscript (page 10):

“...In summary, we have introduced a novel class of highly luminescent TTB_rM-radicals with stable chirality and good photostability. During the submission of this manuscript, we became aware of another study reporting the synthesis of a TTB_rM-Cz radical.^[30] By overcoming the intrinsic low emissivity of TTB_rM, while preserving spin coherence and axial chirality, our work removes a key obstacle...”

[30] K. Nakamura, K. Matsuda, K. Anraku, K. Yamaoka, T. Matsumoto, F. Ishiwari, W. Ota, E. Fujiwara, T. Sato, Y. Inoue, S. Kushida, Y. Yamamoto, T. Hosokai, K. Albrecht, “Luminescent Donor-Acceptor Radical with Propeller Chirality: Bright and Photostable Red Circularly Polarized Luminescence and Whispering Gallery Mode Resonance” *ChemRxiv* **2026**, doi: 10.26434/chemrxiv.10001675/v1.

Referee: 3

Comments:

In this paper, Richert, Kuehne, et al. report on the functionalization of the well-known tris(tribromophenyl)methyl (TTBrM) radical with carbazole units, demonstrating the feasibility of the Buchwald–Hartwig cross-coupling on the parent brominated HTTBrM precursor, on a very interesting organic chemistry synthesis development.

We thank the reviewer for the in-depth analysis, suggestions for improvement, and for recognizing the significance of the synthetic development. We agree that the demonstrated Buchwald–Hartwig functionalization of the HTTBrM precursor represents an important advance. This strategy enables access to donor-functionalized TTBrM radicals, thereby extending the well-established chemistry of TTM-based systems to their brominated analogues. Importantly, this synthetic approach provides a general platform for the incorporation of electron-donating units such as carbazole, opening new opportunities to design and study TTBrM-based radicals, diradicals, and multiradicals.

With a view to assessing the relevance of the resulting TTBrM-Cz radicals as potential spin-based qubits, the spin dynamics have been studied in frozen solutions via standard Hahn-echo and inversion recovery pulse sequences. The performance in terms of phase quantum coherence is one order of magnitude poorer than reported for the parent TTBrM radical (J. Mater. Chem. C, 2024, 12, 5150–5156).

The absorption properties of the TTBrM-Cz radicals were examined and the trends of the changes to the minor bands in the visible region across the series are rationalized in terms of the donor properties of the Cz substituent. DFT calculations help in rationalizing this behavior. These conclusions are supported by the results from cyclic voltammetry. For all the TTBrM-Cz derivatives, the pure enantiomers can be separated by chiral HPLC, and as expected, the optically active species remain stable at room temperature against racemization. Accordingly, CD spectra can be obtained.

In addition, the compounds exhibit photoluminescence (PL) with high quantum yields. On the other hand, circularly polarized PL is extremely poor.

Photostability studies for the radicals have been performed in hexane and the $t_{1/2}$ times have been benchmarked as a function of the number of Me substituents on the carbazole (Cz) unit, with a direct correlation. A considerable increase of $t_{1/2}$ values is observed in toluene. Realizing the functionalization of the HTTBrM closed-shell precursor with carbazoles is interesting, considering that the Cl counterparts have been obtained previously in several instances via cross-coupling reactions (e.g.; J. Mater. Chem. B, 2024, 12, 6840–6846; Mater. Chem. Front., 2017, 1, 2132–2135; Mater. Horiz., 2019, 6, 1265–1270, none of them cited in the submitted draft) leading to interesting optical properties, including enhanced luminescence efficiency. Thus, in the analogous manner as for other aryl-bromides of high complexity (e.g. ACS Omega, 2024, 9, 50446–50457; J. Org. Chem. 2021, 86, 17594–17605), the Buchwald–Hartwig cross-coupling has proven effective for this functionalization. However, it must be emphasized that this was indeed hard to predict, considering the importance of the steric hindrance posed by the Br substituents to the oxidative insertion of Pd(0) into the C-Br bond.

We thank the reviewer for drawing our attention to these very relevant studies. We agree that carbazole-functionalized TTM radicals and related systems have been previously reported and provide important context for the development of donor-functionalized TTBrM emitters. We have now incorporated the suggested references into the Introduction to better reflect the state of the art and to position our work within this broader framework.

Changes to the manuscript (page 2):

To increase ϕ , TTM has been coupled to electron donating substituents, disrupting the C_3 symmetry by producing a charge transfer (CT) excited state,^[10-13] strongly increasing its ϕ to 88 % (see **Figure 1**).^[14,15]

References:

- [11] Anraku, K., Matsuda, K., Miyata, S., Ishii, H., Hosokai, T., Okada, S., ... & Albrecht, K., "A water-soluble luminescent tris (2, 4, 6-trichlorophenyl) methyl radical-carbazole dyad" *Journal of Materials Chemistry B* **2024**, 12(28), 6840-6846.

- [12] Dong, S., Obolda, A., Peng, Q., Zhang, Y., Marder, S., & Li, F., "Multicarbazolyl substituted TTM radicals: red-shift of fluorescence emission with enhanced luminescence efficiency" *Materials Chemistry Frontiers* **2017**, 1(10), 2132-2135.
- [13] Abdurahman, A., Peng, Q., Ablikim, O., Ai, X., & Li, F., "A radical polymer with efficient deep-red luminescence in the condensed state" *Materials Horizons* **2019**, 6(6), 1265-1270.

The resulting TTBBrM-Cz exhibit moderate to poor phase memory times, as revealed from routine fits of the decay of Hahn-echo detected signals, ascribed precisely to the Br substituents and their effect on the SOC. The optical properties of the new Cz derivatives outperform these from the not functionalized radical precursors, as one could have expected from previous work on the Cl analogues (see for example *J. Mater. Chem. B*, 2024, 12, 6840–6846). Therefore, the potential improvements reported here with the new TTBBrM-Cz derivatives do not justify publication in *Nature Communications*. The fact that the racemization in solution is slower with the Br versions than for the Cl ones is hardly relevant, since any implementation of spin-based qubits will take place on solid state hybrid architectures. Indeed, optical spin readout requires fulfilment of harsh conditions, such as the demonstration that the optical transitions couple differently with the spin states encoding a qubit or the ability to polarize the spin upon optical excitation. None of this is addressed in the current paper, which could be considered a real breakthrough.

We thank the reviewer for this detailed and thoughtful assessment. We agree that donor-functionalization of trityl radicals is known to enhance photophysical properties, as demonstrated for chlorinated analogues. It is precisely this expectation that triggered us to investigate this new class of donor-functionalized TTBBrM radicals. However, the key point of our work is that such functionalization has not previously been accessible for the sterically congested TTBBrM scaffold, due to significant synthetic challenges associated with C-Br activation.

Overcoming these constraints represents a central advance of our study. It enables, for the first time, a systematic investigation of donor-functionalized TTBBrM radicals, thereby extending structure-property relationships established for TTM systems to a fundamentally different and more demanding molecular platform.

While the observed phase memory times (T_m) are reduced compared to the parent TTBBrM radical, this behavior provides important insight into the role of heavy-atom substitution and spin-orbit coupling in these systems. In this context, our work offers a direct comparison across TTBBrM, donor-functionalized TTBBrM, and TTM-based analogues, which has not been available to date.

We further emphasize that the novelty of our work does not rely solely on incremental photophysical improvements. Rather, it lies in establishing:

- an efficient and general synthetic strategy to access donor-functionalized TTBBrM radicals with 52% - 58% yield along the two important steps of Buchwald-Hartwig coupling and radicalization (deprotonation, oxidation) for the donor functionalized radicals.
- a previously unavailable class of thermally stable chiral, emissive open-shell systems, and
- a comprehensive structure-property understanding spanning spin dynamics, photophysics, and chiroptical behavior.

We respectfully disagree with the reviewer's assessment that the observed increase in racemization barrier is of limited relevance. By contrast, our work establishes the necessary molecular foundation for future developments of strongly emissive radicals with persistent chirality. By providing access to a previously unattainable class of stable, emissive, and chiral radicals, we open the door to the implementation of advanced strategies – such as supramolecular organization (Meijer & Friend, *Science* 387,1175 (2025)) or chiral doping in LC polymer matrices (Wade & Fuchtnr, *Nat. Photon.* 18, 658 (2024)) – strategies that have recently proven effective in amplifying CPL and enabling spin-dependent optical phenomena.

Finally, while we agree that key requirements for optical spin readout are beyond the scope of the present study, our work establishes the molecular foundation necessary to pursue such directions. In particular, access to stable, donor-functionalized TTBBrM radicals opens opportunities for future

developments, including supramolecular organization and chiral host–guest systems, which have recently been shown to amplify CPL and enable spin-dependent optical effects.

Taken together, we believe that our manuscript represents an enabling advance that expands the chemical space of emissive radical systems and provides a foundation for future exploration of spin-photon functionalities.

I suggest addressing some minor issues before submitting the paper to a more specialized journal:

Minor points:

1. I could not find any indication of the concentration used for the EPR measurements, especially for the relaxation time measurements. It would be relevant to know whether the same concentration was used throughout, and whether it could be considered sufficiently low to neglect intermolecular dipole–dipole contributions to relaxation. Judging from the signal-to-noise ratio, the samples were likely quite dilute, but this should at least be explicitly stated.

We apologize that this important detail was not explicitly mentioned in the text. Based on our own experience, we fully agree with the reviewer that a comparable concentration is especially important for measurements of relaxation times. When recording the presented data, we took extra care to keep the concentration the same for all samples to be compared and at a relatively low value of 0.1 mM. The concentration (0.1 mM throughout) is now stated in the section on EPR sample preparation in the Supporting Information as well as the relevant figure captions.

Changes to the Supporting Information (page 10-11):

“... Continuous wave (cw) electron paramagnetic resonance (EPR) spectroscopy at the X-band (9.75 GHz) was performed on a Bruker EMXNano benchtop EPR spectrometer. The measurements were carried out in degassed liquid toluene solution ($c = 0.10 \text{ mM}$) at room temperature in quartz EPR tubes with an outer diameter of 4.95 mm (inner diameter of 4.2 mm). The recorded spectra were frequency-corrected to 9.75 GHz and field-corrected using a carbon fiber standard with $g = 2.002644$.^[5]

Pulse EPR spectroscopy was performed at the Q-band (34 GHz) on a Bruker ELEXSYS E580 EPR spectrometer equipped with a Bruker EN45107D2 resonator. The measurements were carried out in frozen toluene solution ($c = 0.10 \text{ mM}$) at 80 K using an Oxford Instruments nitrogen gas-flow cryostat (CF 935) and in quartz EPR tubes with an outer diameter of 1.6 mm (inner diameter of 1 mm).

Electron spin echo (ESE) detected field-swept EPR spectra were recorded using the Hahn-echo sequence, $\pi/2 - \tau - \pi - \tau - \text{echo}$, with a 32 ns π -pulse. The spectra were frequency-corrected to 34 GHz and field-corrected using a carbon fiber standard with $g = 2.002644$.^[5]

The T_1 -relaxation data were recorded using a picket-fence sequence composed of 28 saturation pulses (12 ns) and the Hahn-echo detection sequence, using a 32 ns π -pulse and 200 ns inter pulse delay. The data were fit using a biexponential decay function of the form

$$I(t) = A \exp\left(-\left(\frac{t}{T_1}\right)\right) + A' \exp\left(-\left(\frac{t}{T_1}\right)\right) + C.$$

The T_m -relaxation data were recorded using the 2-pulse Hahn-echo sequence, varying the inter pulse delay τ . The data were fit using a stretched monoexponential decay function of the form

$$I(2t) = A \exp\left(-\left(\frac{2t}{T_m}\right)^\beta\right) + C.$$

While the fitting with a stretched exponential function reproduces the experimental data well, it complicates the direct comparison of the extracted relaxation times T_m , which is why both T_m and β parameters are given.

All samples were left to stabilize for two hours at the measurement temperature of 80 K before the relaxation data were recorded.

Comparative T_m measurements in deuterated frozen toluene solution were carried out analogously, whereby a slightly higher concentration of $c = 0.16$ mM was used for the reference compound **TTM-Cz**.

...

2. The variation in phase memory time among the various Cz derivatives is very small (from ~ 1.5 μ s to ~ 1.7 μ s). The trend observed could easily change because of many possible sources of experimental error, for example, differences in glass formation of the toluene matrix, which are inherently difficult to reproduce perfectly. Since the observed trend is surprising and counter-intuitive, in-depth study and confidence intervals are necessary before any hypotheses are put forward. Certainly, the explanation given; **“screening of spins from the decohering environment”**, seems highly questionable. Methyl groups themselves are a source of relaxation, so it is difficult to imagine that a frozen toluene matrix contributes more to relaxation than the methyl groups of the molecule. This could, however, be tested straightforwardly by using deuterated toluene, where the contribution of solvent nuclear spin diffusion should be significantly reduced.

We thank the reviewer for pointing this out and agree that the statement **“screening of spins from the decohering environment”** was not appropriate in this context. The statement was now removed and the narrative adapted.

We also agree that the measurement of relaxation times is prone to errors and that many experimental parameters may influence the outcome of the measurement, compromising the comparability of the results. Apart from differences in the glass formation, which we believe is not the main source of error, the sample concentration and, in particular, the time that the sample is left to stabilize at the measurement temperature prior to the relaxation measurements play a major role.

Aware of these sources of error, we maintained identical concentrations for all samples and left the samples to stabilize for the same amount of time (approximately 2 hours) before recording the relaxation data. We estimate the error on the T_m values to be of the order of ± 0.1 μ s, partially also due to the use of stretched exponentials in the fitting procedure. It is well known that T_m times extracted from fits with different stretch factors β are not directly comparable. Since no good fit could be obtained when keeping the stretch factor the same for all samples, we now quote these factors in the Supporting Information (see **Figures S4** and **S5**).

Regarding differences in the stretch factors, we would like to note that a lower value was obtained for sample **TBrM-Cz** as compared to the two methylated analogues. This may partially explain the lower T_m value obtained from the fit for **TBrM-Cz** and suggests that the three samples might be more similar in their relaxation behavior than inferred from the fitted T_m values. By eye the three decays are very comparable, indeed. As a consequence, we now refrain from any interpretation of the differences observed. However, we believe that it can be stated with confidence that the addition of methyl groups on the carbazole moiety, contrary to expectations, does not seem to have a marked negative impact on the spin coherence times.

Following the suggestion of the reviewer, we now also included complementary relaxation data recorded in deuterated frozen toluene in the Supporting Information. As expected, the measured T_m values increase (from roughly $1.6 \pm 0.1 \mu\text{s}$ to $2.1 \pm 0.1 \mu\text{s}$). Interestingly, the increase is the same for the **TBrM-Cz** and **TBrM-Me₂Cz**. This underlines the observation that these methyl groups do not cause additional spin decoherence. It is certainly true that both internal and matrix protons contribute to spin decoherence in the present cases.

Changes to the manuscript (page 5-6):

“... Previous studies have shown that the **TBrM** radical exhibits significantly extended phase memory times T_m compared to **TTM**, rendering **TBrM** interesting as a molecular qubit.^[9] To determine T_m and the spin-lattice relaxation time T_1 , we perform pulse EPR measurements in frozen toluene solution at 80 K (see **Figure 3b** and **Figures S2-S4** in the SI). We are able to reproduce the reported T_1 for **TBrM**, obtaining a value of 0.05 ms (literature: $T_1 = 0.07$ ms) (see **Figure S3**). However, the measured phase memory time $T_m = 1.5 \pm 0.1 \mu\text{s}$ is significantly shorter than the reported literature value ($T_m = 17.0 \mu\text{s}$) (see **Figure S4**).^[9] For comparison, the corresponding relaxation times that we determine for **TTM** ($T_1 = 0.80$ ms; $T_m = 4.0 \pm 0.1 \mu\text{s}$) and **TTM-Cz** ($T_1 = 1.05$ ms; $T_m = 4.1 \pm 0.1 \mu\text{s}$) are markedly longer than those of **TBrM**. The overall reduction in spin coherence is not unexpected for brominated radicals relative to their chlorinated analogues. This can be attributed to the higher atomic mass of bromine, which enhances SOC and thus increases the rate of spin relaxation, leading to shorter phase memory times. The discrepancy between our measured and the reported T_m values for **TBrM** may arise from solvent effects, variations in temperature and measurement conditions, or from differences in sample purity.

When we functionalize the **TBrM** with our three different **Cz**-based electron donors, we observe slightly increased T_1 values compared to the unsubstituted **TBrM** with $T_1 = 0.07$ ms, 0.09 ms, and 0.10 ms, respectively for **TBrM-Cz**, **TBrM-MeCz**, and **TBrM-Me₂Cz** (see **Table 1** and **Figure S3**). No significant variation in T_m is observed with **Cz** functionalization and with successive methylation of the **Cz**-donor, with T_m values ranging between $1.5 \pm 0.1 \mu\text{s}$ and $1.6 \pm 0.1 \mu\text{s}$ (see **Figure S4**). These coherence times can be slightly increased when measuring in deuterated toluene, with $T_m \sim 2.1 \mu\text{s}$, while T_m of the **TTM-Cz** reference is increased to $4.7 \pm 0.1 \mu\text{s}$ (see **Figure S5** in the SI). The results

demonstrate that methylation of the carbazole does not impact negatively on the phase memory times.

...

Changes to the Supporting Information (page 11, Figures S4-5):

Figure S4: Spin coherence decay data in frozen toluene solution ($c = 0.1 \text{ mM}$) with a stretched exponential decay fitted to the experimental data to determine T_m .

Figure S5: Spin coherence decay data in deuterated frozen toluene solution ($c = 0.1 \text{ mM}$, 0.16 mM for **TTM-Cz**) with a stretched exponential decay fitted to the experimental data to determine T_m .

3. The CPL performance is very poor and heavily affected by very low signal-to-noise ratio. In fact, the Gauss fits as guide to the eye in figures d, e and f (especially the first two) are rather unconvincing and put into question the quantitative glum provided.

We thank the reviewer for raising this important point regarding the signal-to-noise ratio in the CPL measurements. We agree that the CPL signals are relatively weak and exhibit a limited signal-to-noise ratio. However, this is inherent to systems with g_{lum} values of the order of 10^{-4} , as observed for our TTBBrM-Cz derivatives. Such values are typical for small chiral organic molecules in solution and are therefore expected.

To ensure the reliability of our analysis, we based our quantitative evaluation of g_{lum} on the data provided in the Supporting Information (Figure S16), where the g_{lum} spectra are less affected by processing and noise. The extracted g_{lum} values are reproducible and consistent across independent measurements.

Importantly, the obtained g_{lum} values are in line with those reported for related systems, including the parent TTBrM radical and established chlorinated analogues such as TTM and TTM-Cz, which also exhibit g_{lum} values of the order of 10^{-4} . In this context, significantly larger values would not be expected for this class of compounds.

For clarity, we note that the Gaussian fits in Figures d–f are intended solely as visual guides and are not used for the quantitative determination of g_{lum} .

Referee: 4

Comments:

We thank the reviewer for taking the time to co-review our manuscript.

REVIEWER REPORT(S):

Referee: 1

Comments:

The authors have properly addressed the proposed issues, and the manuscript is recommended for acceptance.

Referee: 3

Comments:

After reading the revisions and explanations given by the authors in their letter, I recognize that they have made important efforts to address the referee's concerns. However, the paper brings essentially the same message as the initial one. I have a very similar opinion as that of Referee #2. The interesting organic synthesis development does not justify publication in Nature Communications. As mentioned by the authors, the enhanced photophysical properties of the Br derivatives were already established on the Cl analogues, while the spin dynamics are not really better. The improvement on the stabilization of the optical activity is expected, as has been observed before (Chem. - A Eur. J. 2020, 26, 3776–3781). In my opinion, this interesting paper, in its form after the revisions, should be transferred to a more specialized journal.

In fact, it is not expected that the spin properties differ from those of the previously published TTB_rM parent radical upon Cz attachment. (The quoted reference describes the parent radicals, which are effectively dark.) The TTB_rM-Cz derivatives reported here have so far been inaccessible. We now provide access and evidence that this new class of molecules exhibits PLQY comparable to that of the related TTM-Cz systems. This opens up a new realm of light-emitting trityl radicals.

Diradicals based on TTM and TTB_rM moieties are now accessible, together with stable propeller chirality. Our results open up the possibility to design new constructs and assemblies with more pronounced CPL characteristics. Therefore, this work presents a first step into this new class of strongly light-emitting TTB_rM-type radicals.

Referee: 4

Comments:

We thank the reviewers for the time and expertise to review our manuscript. The direct and constructive criticism has helped us to strengthen the manuscript and some of our hypotheses. As a result, we believe the revised version provides a more robust and coherent contribution.